# Sociocultural practices, beliefs, and myths surrounding newborn cord care in Bayelsa State, Nigeria: A qualitative study

Chika O. Duru[1]*, Abisoye S. Oyeyemi[2], Adedotun D. Adesina [3], Ijeoma Nduka[4], Charles Tobin-West[5], Alice Nte[6]

**1** Department of Paediatrics and Child Health, Niger Delta University, Yenagoa, Bayelsa State, Nigeria,
**2** Department of Community Medicine, Niger Delta University, Yenagoa, Bayelsa State, Nigeria,
**3** Department of Medical Services, Nigerian Law School, Yenagoa Campus, Agudama, Yenagoa, Bayelsa State, Nigeria, **4** Department of Community Medicine, Abia State University Teaching Hospital, Uturu, Abia State, Nigeria, **5** School of Public Health, University of Port Harcourt, Choba, Rivers State, Nigeria,
**6** Department of Paediatrics, University of Port Harcourt Teaching Hospital, Port Harcourt, Nigeria

* duru_chika@yahoo.com

**Data Availability Statement:** All relevant data are within the paper and its Supporting information files.

## Abstract

Persisting sociocultural beliefs have continued to significantly influence the adoption of recommended newborn care practices by women in Sub-Saharan Africa. This study aimed at identifying the sociocultural practices, beliefs, and myths surrounding newborn cord care by women residing in Bayelsa State, Nigeria. This was a qualitative study that involved 24 women and 3 traditional birth attendants (TBAs) in three focus group discussions and three in-depth interviews respectively. Interview guides were used to lead the discussions and the interviews which were audiotaped, translated and then transcribed. Thematic analysis was done using NVivo QSR version 12.2 Pro. Several themes describing various sociocultural practices, beliefs, and myths surrounding cord care were uncovered. Most women preferred to be delivered by a TBA who usually cuts the infant's cord with a razor blade and ties the stump with hair or sewing thread. Substances used for cord care included methylated spirirt, "*African never-die*" leaf, and "*Close-Up*" toothpaste. All the participants agreed that methylated spirit was a potent antiseptic for cord care but none of them had heard about nor used chlorhexidine gel. It was a common belief that abdominal massage and the application of substances to the cord were solutions to common cord-related problems. Mothers, TBAs and relatives were influential regarding choices of cord care practices. Sociocultural practices, beliefs, and myths are still major barriers to the adoption of recommended cord care practices by women in Bayelsa State. Interventions should be targeted at improving delivery in health facilities and educating women in the community on good cord care practices.

## Introduction

Cultural influences, beliefs, and practices in the care of children are an age-long phenomenon in many parts of Africa [1, 2]. Most have been in existence before the introduction of orthodox

**Funding:** The authors received no specific funding for this work.

**Competing interests:** The authors have declared that no competing interests exist.

medicine and are still being held and practised by many tribes and families [3]. Most of these traditional beliefs transcend generations and are difficult to change. In Nigeria and several countries in Sub-Saharan Africa, harmful and non-beneficial cord care practices abound and these are influenced by various sociocultural beliefs surrounding the care of the cord which have been passed down from generation to generation [4–7]. These traditional practices are increasingly being reported especially in rural settings where home delivery is common [7, 8]. The application of various substances to the freshly severed cord is a common traditional practice in sub-Saharan Africa and is acceptable to many mothers because of their 'perceived' ability to aid rapid healing and drying of the cord [4–7, 9–13]. In rural communities in Uganda, the newborn period was regarded by members of the community as the period between the 'birth of a child and the time of cord separation'[14]. Thus, shortening of the period of cord separation was a desired reason for the adoption of some cord care practices especially those that hastened the drying of the cord [15, 16]. The separation of the infant's cord was considered an important milestone of newborn care as it signaled the end of confinement for the mother after delivery [15, 16]. In several African settings, cutting of the newborn cord with unsterile instruments is another common cultural practice, especially during home deliveries [7, 17, 18]. Instead of the recommended use of sterile blades usually obtained from the sterile delivery pack, some traditional birth attendants (TBAs) use unclean blades or other materials to cut the newborn cord thus exposing the child to the risk of infections. In Bayelsa State and other states in the Niger Delta region of Nigeria, materials such as "hair thread", palm fronds or bamboo sticks are commonly used to cut the umbilical cord as it was believed that using a sharp object would delay cord separation [18]. The use of hot fomentation on the cord has also been reported as a common practice in Africa and Asia. The use of heat was thought to promote faster cord healing and remove the "bad" smell from the cord [11, 13, 17, 18]. Some Nigerian studies reported that heat was applied to "sterilize" the cord before the application of either an antiseptic or other substances; a harmful practice which can expose the child to burn injuries [4, 9].

The adoption of these practices by mothers has been found to be harmful as they predispose the neonates to infections such as omphalitis and neonatal tetanus, which are both important causes of neonatal mortality [19–23]. According to the 2018 National Demographic and Health Survey (NDHS), despite the fact that up to 62% of women had received sufficient doses of tetanus toxoid vaccine to prevent neonatal tetanus, cases of neonatal tetanus have still been reported due to poor hygienic practices [24]. The fact that up to 59% of women in Nigeria deliver their babies at home without the assistance of a Skilled birth attendant also exposes them to unhygienic practices endorsed by traditional birth attendants [18, 24].

Bayelsa State has one of the poorest child health indices in Nigeria according to the most recent National Demographic and Health Survey (2018 NDHS) [24]. Continuing harmful cultural practices that predispose children to infections and death are partly responsible for these poor indices. Several Nigerian studies on cord care have described the various cord care practices adopted by mothers but there has not been any that explored the underlying reasons for their adoption. This study was thus conducted to describe the various types of cord care practices employed by women in Bayelsa State and to explore the cultural beliefs and myths facilitating their adoption. We hope that findings from this study would encourage the adoption and implementation of extant policies on optimal cord care and inform behaviour change communication programmes that will promote healthy cord care practices in Bayelsa State. The study used a mixed-method approach but this paper presents findings from the qualitative component of the study.

## Materials and methods

### Study design

This was the qualitative study component of a mixed method study carried out in Bayelsa State, Nigeria over a two- month period; 1st May to 30th June 2021.

### Study area

This study was conducted in Bayelsa State, one of the southernmost States of Nigeria with an estimated 2021 population of 2,633,466 projected from the 2006 National Census [25, 26]. The State is made up of eight local government areas (LGAs) with 105 wards and its capital, Yenagoa, is located in Yenagoa LGA. The major local language of Bayelsans is Ijaw (Izon) though other local languages such as Nembe, Ogbia and Epie-Atissa, are also widely spoken in the State. Bayelsa State has one of the largest crude oil and natural gas deposits in Nigeria and is also one of the largest producers of oil and gas products in the country. Despite this, Yenagoa Local Government Area is the only urban LGA while the rest of the State is rural with most of the inhabitants being mainly civil servants, fishermen and farmers.

### Sampling technique

The participants for the study were selected using a multistage probability sampling technique. This involved the selection of one local government area from each of the three senatorial districts in Bayelsa State by balloting. Thereafter one ward was randomly selected from each LGA and then one community was selected from each ward by balloting. The participants for the qualitative interviews were purposively selected from the three communities; eight women and one TBA per community for the focus group discussion (FGD) and in-depth interview (IDI) respectively. After community entry and explanation of the purpose of the research, the women leader of each community was identified and she helped to gather eligible women who were vocal and willing to discuss the research subject at the FGD. For the IDI, the most popular TBA in each community was identified and interviewed. Thus one FGD and one IDI were conducted in each senatorial district giving a total of three FGDs and three IDIs. A traditional birth attendant was defined as a person who usually assists women during pregnancy and delivery but who has no formal training to do so. Data was collected using FGD/in-depth interview guide, an audiotape and a notebook. The interview guide was made up of 7 main questions and 14 probes questions exploring different areas of cord care practice immediately after birth and during early neonatal life.

The interviews were conducted in English and *pidgin* English by the principal researcher while a research assistant recorded the sessions with an audiotape and took notes. *Pidgin* English is the lingua Franca in Bayelsa State and most women were comfortable with it. Each session of the FGD lasted between 30 to 97 minutes and that of the IDI between 45 and 60 minutes and the interviews were stopped once saturation was reached. The recordings in *pidgin* English were translated to English language by a local interpreter and then all the recordings were transcribed and word-processed. Participants' anonymity in the interviews was protected as the participants were represented with the letter "R", which denotes a particular but unidentified participant.

### Data analysis

Thematic analysis, through the inductive model, was done using NVivo QSR version 12.2 Pro. Emerging themes from the transcribed interviews and discussions were coded. As themes were generated from the interviews, sections, and sub-sections were also developed. Related

themes were grouped hierarchically and thereafter the identified themes were pooled together to summarize the study findings.

## Ethical consideration

Ethical approval for the study was granted by the Research Ethics Committee of the University of Port Harcourt, Rivers State (UPH/CEREMAD/REC/MM74/059). Permission to conduct the study in each community was granted by the respective community heads. Written informed consent was obtained from all eligible participants before the commencement of the study. All information obtained from the participants was treated as confidential.

## Results

### Sociodemographic characteristics of participants in the focus group discussions and in-depth interviews

**The in-depth interview participants.** The three traditional birth attendants who took part in the in-depth interviews comprised two women (66.7%) and one man (33.3%) with ages 53, 60 and 72 years respectively. None of them had a formal education. They were all primarily farmers and usually assisted women during pregnancy and delivery.

**The focal group discussants.** A total of 24 women participated in the FGDs. Their ages ranged from 19 years to 51 years with a mean age of 35.4 ± 5.9 years. Majority (62.5%) were married, 62.5% had a secondary level of education and slightly above a third of them (37.5%) were unemployed.

### Themes identified during the focus group discussions and in-depth interviews

During the 3 Focus Group Discussions and 3 In-depth Interviews the following themes were identified:

**1. Preferred places of delivery and reasons for their choices.** More than half of the mothers interviewed preferred to deliver at the homes of traditional birth attendants (TBAs), popularly referred to as "massaging homes" in these communities. Less than half preferred health facilities- "health centres" as their places of choice to give birth. According to some of the FGD participants:

*"Mama Ijaw place, that is massaging place"<R3 FGD_Yenagoa LGA>*

S3 Text

*"Most women prefer to deliver in native place".<R2 FGD_Ogbia LGA>*

S1 Text

*"I always give birth in the massaging place" < R1, FGD_Ogbia LGA>*

S1 Text

*"As for me, it's the nurse that delivers the baby, but it's at home" < R2, FGD_Ogbia LGA>*

S1 Text

The reasons mentioned for this preference by most of the respondents was fear of caesarian section, distance to the home, cost and level of care received. As some of the participants summed up:

*"...If you go to the hospital or health centre, the doctors will insist on an operation.. ..so instead, some people go to the massaging home. Money is also involved, but some people go because it is nearby... it depends on when the labour comes, so some people prefer to go to the nearby place". <R2, FGD_Sagbama LGA>*

S2 Text

The TBAs who own some delivery homes reported high service delivery records at their various TBA homes. As two of them explained,

*"I have delivered up to four women this week...but some weeks, I can attend to up to ten women....."<TBA, IDI_Yenagoa LGA>*

S6 Text

*"Yes, I attend to a lot of births in this community"<TBA, IDI_Ogbia LGA>*

S4 Text

**2. Cultural myths or beliefs about the newborn's cord.** Cultural myths or beliefs about newborn's cord abound in Bayelsa State. Almost all the participants affirmed the existence of some cultural beliefs and practices in their areas. Common cultural myths and beliefs mentioned included the beliefs that it was a taboo for the mother to see her child's umbilical cord;

*"Because in many places, the cord is forbidden...there are some people who are not supposed to see the child's cord because when they see it, the baby will die..." < R8, FGD_Ogbia LGA>*

S1 Text

*"Basically, the tradition here is that during the first seven days when the child is at home, the mother doesn't come outside...if she comes outside, she may meet people that may use charms "black magic"(charms) on the child when the navel (cord) has not fallen... and then the child will fall sick... <R2,FGD_Sagbama LGA>*

S2 Text

*"Yes, there are people that are not supposed to see the cord...one woman lost her seven children because of this..., during the delivery of her eighth child they covered her eyes so that she will not see the baby and somebody took that child away from her for two weeks then they brought the child back to her. That's the only child that she has..."<TBA, IDI_Yenagoa LGA>*

S6 Text

**3. Cord-cutting and tying practices for newborns by women in the community.** Almost all the participants knew about the use of a clean razor blade for cutting the newborn's

cord. The razor blade was the most frequently mentioned cord-cutting device used, followed by a pair of scissors and raffia palms. However, a few women reported using local herbs in cutting the newborn's umbilical cord. A few participants responded as follows:

> *"Most people use the native leaf to cut their baby's cord; it is the 'African never die'"*. . . **because its blade is very sharp.. . . < R1, FGD_Ogbia LGA>**

S1 Text

> *"They use a blade to cut it; they will drag the navel (cord) small before they cut it"* <R1, FGD_Yenagoa LGA>

S3 Text

For cord tying, about half of the participants mentioned that black hair thread was the most commonly used, and some others mentioned that cord clamp/clips were used sometimes. Most participants reported that tying and cutting of umbilical cords were performed by TBAs. However, a few noted that healthcare workers did the cutting. No special ritual or cultural practices during the cutting of cords was reported as pointed out by some participants:

> *"They will use black thread to tie the cord before they will use a blade to cut"* <R4, FGD_Sagbama LGA>

S2 Text

> *"If it is in the clinic, they will sterilize it.. . . they will boil it before they use it on the baby's body"* <R1, FGD_Yenagoa LGA>

S3 Text

> *"They use raffia palm fronts to tie the cord"* < R3, FGD_Ogbia LGA>

S1 Text

Findings from the in-depth interviews corroborated what the discussants stated. All of them pointed out that they used "hair thread" and blade for tying and cutting the cord. Some key informants noted,

> *"After I have delivered the child. . . I will just draw the cord to make the dirt come out from inside and then use a clean rope to tie it. I will use a new blade that has never been used on anybody else to cut it. . .<TBA, IDI_Sagbama LGA>*

S5 Text

> *"I use the normal thread that they use to sew clothes or tie (plait) hair. . . clean ones. Then between the stomach and the cord, I will measure two or two and a half-inch then I will tie it and cut. No other special thing is done before the cutting except that I tie it two places so that blood will not pour out from the cord and if one end gets loosened, the other one will still hold the cord"* <TBA, IDI_Yenagoa LGA>

S6 Text

**4. Cord care practices immediately after birth.** Participants listed several unique methods of caring for the newborn's umbilical cord. Many of the methods have their roots in the culture and belief systems of the communities. Some of these methods include the use of Methylated spirit, herbal leaf called 'never die leaf', cooking pepper, alligator pepper, ash, "Close-Up" toothpaste, saccharin powder, ointment ("ROBB"), breast milk and hot water. According to some of the participants,

*"I use (methylated) spirit and cotton wool till the cord falls. . .usually in a week's time. Sometimes it depends on the cord, if it is thick, it will take a longer time. . .but I still use the (methylated) spirit just to clean every time till the cord heals completely.. . ." <R8, FGD_Yenagoa LGA>*

S3 Text

*"First of all, we use (methylated) spirit to clean the navel(cord) so that it will not have an odour, then we use salt (cooking salt), "Never die" leaf, and ashes(from firewood) to finish it". <R1, FGD_Ogbia LGA>*

S1 Text

*"I use "Close–Up" toothpaste which is the best. . ...after two or three days, the cord will fall off". < R2, FGD_Ogbia LGA>*

S1 Text

*"After cutting the cord, at home we use hot water, to press it so that the heat will go inside the belly of the child, after we use (methylated) spirit to clean it and then, after that we apply "ROBB" and then "alligator pepper" so that the hotness of the pepper will enter inside the stomach. . . that process will not allow the child's cord to be painful or have any issues.."< R4, FGD_Ogbia LGA>*

S1 Text

*"After the cord has fallen, I press it with hot water. I put breast milk, and I also chew "alligator pepper" and put it inside. I also I put "never-die" leaves on fire and press the water (extract) inside" < R4, FGD_Sagbama LGA>*

S2 Text

*"It is my mother-in-law that baths the baby; after bathing the baby, she will put "dusting" (menthol) powder inside a tissue paper and wrap the navel (cord) with a bandage till the cord cuts after seven days. . . if the cord is making some kind of noise, she will use hot water to press the navel or she can apply "alligator pepper".. . . < R5, FGD_Yenagoa LGA>*

S3 Text

Similar practices with slight variations were also reported by the participants in the in-depth interviews, the TBAs, and some of them put it thus:

*"We use hot water, and the native leaf 'never die', we grind it and put it on the cord, after at least three to four days the cord will cut.. . .after then we apply (methylated) spirit.*

*When the cord has cut, we clean it with hot water then put the "alligator" pepper inside the cord. Some people will then cover the cord so that air will not enter the body through the cord.,<TBA, IDI_Ogbia LGA>*

S4 Text

*"After we cut the cord, we use hot water and (methylated) spirit to clean it, then we put breast milk on the cord, we also put the native leaf, "Never die" and "alligator" pepper so that the cord will not pain the baby. We put the "never die" leaves in fire squeeze it, and then put it(the extract) on the cord till after a very short while, the cord will fall." <TBA, IDI_Sagbama LGA>*

S5 Text

*"When somebody delivers in the morning, I only clean the baby. In the evening, I use methylated spirit to clean the cord, then I will apply the "Close-up" (toothpaste) around the navel (cord) for three days.<TBA, IDI_Yenagoa LGA>*

S6 Text

The common reasons for adopting these treatment procedures included the perceived ease and quick cutting-off of the cord, quick healing and drying and reduced abdominal pain. Some other reasons were their perceived ability to prevent infection, bad odour and safety. The following statements summed up these claims:

*"To avoid sickness, to avoid contracting infection" < R1, FGD_Ogbia LGA>*

S1 Text

*"So that the cord will not be painful and also to avoid bad odour". <R2, FGD_Ogbia LGA>*

S1 Text

*"The reason behind it was that I learned that it was safe and it will enable the cord to fall off quickly" < R1, FGD_Sagbama LGA>*

S2 Text

*"I use (cooking) salt with "Close Up" toothpaste so that the cord will dry fast and not smell nor "become rotten". I also put pepper inside water, then put the water inside the cord so that the cord will not pain the baby. Then I put breast milk on the cord so that the cord will dry and not smell" < R3, FGD_Ogbia LGA>*

S1 Text

The traditional birth attendants during the in-depth interviews corroborated these responses with the following statements;

*"'never die' (native leaf) makes the cord cut fast. We start applying "alligator pepper" to the stump after the cord has fallen so that air will not enter inside the baby's tummy (stomach) and the child will not have stomach pain" <TBA, IDI_Ogbia LGA>*

S4 Text

*"We use "ROBB" (hot ointment) and pepper to prevent the cord from being painful. The (methylated) spirit makes the cord to fall quickly because it dries the cord, so in a short while it will cut" <TBA, IDI_Sagbama LGA>*

S5 Text

*"The (methylated) spirit is used to remove germs……after that we use the "Close-up", (toothpaste) to cut it". <TBA, IDI_Yenagoa LGA>*

S6 Text

When asked about any inherent side effect of the methods used, most participants said there were none, while a few indicated some side effects. According to those who suggested some side effects associated with the procedures, below are some statements corroborating this:

*"For my first baby, I used "Close- up" (toothpaste) and it(the umbilical cord) fell off quickly but before I knew it "air had entered the baby's stomach" because the cord was not dry, I regretted using it, that's why I decided to be using (methylated) spirit and "ROBB" (ointment) only" < R8, FGD_Yenagoa LGA>*

S3 Text

**5. Cord problems ever experienced or heard about and solutions.**   More than half of the participants reported having experienced or heard about some cord-related problems within their environment. The problems mentioned were bleeding from the cord, "bad odour", "stomach bloating", fussiness, grunting, and swelling/redness around the base of the cord. The statements below sum up their observations:

*"I have heard stories of children dying….. maybe the cord will start smelling, and they will have to take the child to one place they call NICU(Neonatal Intensive Care Unit) in the hospital…. some people even stay there with their babies for three to four months, so it is a very terrible situation, those are the things I have heard that happen because of the use of those other methods apart from (methylated) spirit" < R1, FGD_Yenagoa LGA>*

S3 Text

*Like one woman, my neighbour, she did not take care of her child's cord well, and she did not even use (methylated) spirit, so the thing (cord) just cut off. Before we knew what was happening, the tummy (stomach) started swelling like, as "if air entered into the baby's stomach" and the baby was not able to breathe, so they took the child to the hospital, but the following day the child died. The cord was also very dark, and it was smelling" < R2, FGD_Yenagoa LGA>*

S3 Text

Almost all the participants believed that such problems were not common in their community. Most participants believed that these cord-related problems resulted from improper care and short cutting of the navel. Consequently, they suggested that these problems could lead to infection or death of affected children.

Concerning the measures to solve these problems, the majority recommended taking the child to 'massaging homes' i. e. TBA homes, while a few suggested the hospital as the best point of care. Participants also described how such problems were solved. According to them,

*"We, as native people, have many ways of doing it. . ...when the baby's "tummy is rumbling", you know that the inside has not healed. So I will mix one native medicine leaf with hot water, then I will use a manual pump and put it through the anus, and push the fluid into the baby's belly (stomach)(an enema). The baby will pass large amounts of green stools which is the cause of the problem. Once that happens the baby will be okay.. . ." <TBA, IDI_Yenagoa LGA>*

S6 Text

*"Some people use native medicine to rob on the cord and then put 'kaikai' (native dry gin). . . this is so that the cord will not pain the baby." < R4, FGD_Sagbama LGA>*

S2 Text

*"You carry the baby to the hospital when you can't handle it by yourself because you don't know anything concerning the sickness" <TBA, IDI_Ogbia LGA>*

S4 Text

*"After I massage the baby, after three days of spraying the medicine, the problem is gone, and the baby will be healed" <TBA, IDI_Sagbama LGA>*

S5 Text

**6. Sources of knowledge of cord care practices.** The sources of knowledge on the use of the various methods for cord care by the women varied. More than half reported that they were taught by their mothers, some by health care workers and others by neighbours, mothers-in-law, TBAs, siblings.and friends.

*"The (use of methylated) spirit I learned from the hospital but the (use of) 'Close- Up' (toothpaste) my neighbour told me to use it so that the cord will cut fast and she said used it before, that is why I used it" < R1, FGD_Sagbama LGA>*

S2 Text

*"The (knowledge about the use of methylated) spirit was (obtained) from the hospital then (use of) the "alligator" pepper (was learned) from my mother" <R6, FGD_Yenagoa LGA>*

S3 Text

**7. Knowledge and use of recommended antiseptics for cord care.** The participants were asked about their knowledge and use of the common recommended antiseptics (methylated spirit or chlorhexidine gel) that were usually used to care for infants' cords. All participants were aware and mentioned methylated spirit as one of the antiseptics. However, none mentioned chlorhexidine gel as an example of recommended antiseptics. Despite the probe for 'antiseptics' used, more than one-third still mentioned the "never die" native leaf as one of the

antiseptics. Others mentioned the use of methylated spirit with either saccharin, dusting powder and breast milk as "potent" antiseptics. All the participants affirmed that methylated spirit was the most commonly used antiseptic however, few opined that methylated spirit was mainly used for cord drying and infection prevention. Some others mentioned that methylated spirit was used for cord cleaning, odour prevention, and to aid quick separation of the cord and it had no known side effects. According to the participants,

*"The (methylated) spirit is used to remove germs and prevent the baby from having tetanus.. . ."* <TBA, IDI_Yenagoa LGA>

S6 Text

*"First of all, we use (methylated) spirit to clean the navel(cord) so that it will not have an odour, then we use salt (cooking salt), "Never die" leaf, and ashes(from firewood) to finish it".* <R1, FGD_Ogbia LGA>

S1 Text

*"No mother has reported any side effects from methylated spirit."* <TBA, IDI_Sagbama LGA>

S5 Text

## Discussion

Inappropriate cord care practices can lead to infection of the umbilical cord stump which may eventually lead to avoidable neonatal death. In this study, we explored, identified and discussed sociocultural practices, beliefs, and myths held by caregivers and traditional birth attendants as regards newborn cord care in Bayelsa State.

The Traditional birth attendant's (TBA) homes were found to be the preferred places of delivery by most of the respondents and the TBAs affirmed this by boasting of their patronage. The popular choice of delivery at the TBA's homes has been similarly reported by other authors [24, 27, 28]. In fact, the 2018 NDHS reported that just a quarter of deliveries (25.1%) in Bayelsa State were attended by a skilled birth attendant (SBA) and fewer actually took place in a health facility, the lowest in the South-south zone of Nigeria [24]. Many reasons have been given for the preference for TBAs over a health facility which include the perceived efficacy of their traditional medicines, lower cost of services and easy accessibility and cultural acceptability compared to the health facility [27, 28]. Efforts should be intensified to address the factors that make the TBAs more appealing so that skilled birth attendance can progressively increase and mother and child can have better care with good outcomes.

Women resident in Bayelsa State also poorly attended antenatal care (ANC) by skilled providers and were the worst in the south- south zone as shown in the 2018 NDHS [24]. Since the majority of women did not attend ANC in health facilities, it is not surprising that a lot of them held on to various inimical cultural beliefs surrounding cord care which is probably due to the information received during contact with the TBAs during antenatal care [29]. Common cultural myths and beliefs mentioned included the belief that it is a taboo for the mother to see her child's umbilical cord. Similar reports of anxiety about seeing and touching the newborn's umbilical cord have been reported among women in Uganda [30] where the cord was thought to resemble a piece of an "intestine" hence the fear that if touched, the child would die. At ANC visits, pregnant women are usually educated about various topics that concern

their health. It is also a time when myths and misconceptions about pregnancy, delivery and early child care practices should be dispelled. Interventions that will promote ANC attendance will be helpful to improve the use of health facilities for antenatal care.

The use of clean cutting agents like razor blades for cutting the newborn's cord which was common among the women in the State was similar to the practice in other places like Uganda [30]. One clean razor blade per child is a welcome practice that should be promoted. The thread used to tie the cord after cutting may however be a source of infection as there is no assurance of it being sterile, given the typical TBAs home setting.

Although methylated spirit, a common antiseptic, was popularly used for cord care by a majority of the respondents, notably it was often used in combination with other substances. The major reason for adopting cord care practices that involved the use of substances like "Close-Up" toothpaste, ash, "never die" leaf, and "alligator pepper" was the need to ensure quick cord separation. These substances were supposedly used as drying agents after the application of methylated spirit to sterilize the stump. Most of the respondents believed that it was important for the cord to fall off early and the stump heal fast and the drying caused by these substances would help achieve this. This, they believe would prevent "bad air" from entering the baby's stomach and subsequently prevent the cord from having a bad odour and becoming infected. Women residing in other parts in Nigeria and Uganda similarly believed that the cord was a vulnerable point through which various kinds of sicknesses could affect the baby and lead to death [14, 30]. The erroneous belief around the use of these substances might account for the relatively high incidence of neonatal sepsis and tetanus still experienced in this locality despite antenatal vaccination given to mother during pregnancy. It is therefore important to care for the cord appropriately to prevent diseases and death that could occur due to improper cord care.

Participants in this study had at one time experienced problems due to improper cord care or knew someone who had. Experiences reported include bleeding from the cord, "bad odour", "stomach bloating", fussiness, grunting, and swelling/redness around the base of the cord. Some participants mentioned death of children as one of the complications from improper care of baby's cord. The complications of improper cord care experienced by the participants of this study are similar to those reported among mothers in Calabar in Cross River State, Nigeria by Osuchukwu, *et al* in 2018 [31]. The women in that study reported symptoms such as fever, red skin around base of cord and foul smell due to cord infection. Other studies reported high proportion of cord infections among neonates who used native topical preparations and other unhygienic materials on the umbilical stump [17, 32, 33].

On management of cord infections, mothers in this study had varying views on treatment. Some believed hospitals were a good option while others felt the use of herbal remedies would take care of any cord-related infections. Concerning the management of cord infections, our findings are similar to those of Osuchukwu, et al [31] which showed that only a few caregivers visited the hospital when the cords were infected while the majority managed the infections at home with herbs. Muella and Johnson [34] found that only few mothers treated cord infections in the hospital as against the numerous others that preferred to manage their babies at home. This is consistent with findings from this study. Adequate knowledge and its application in the care of the newborn are key to the reduction of child morbidity and mortality.

The sources of knowledge of cord care practices reported in this study are similar to those reported by Nduka & Nduka [2] where mothers reported caring for their infants from experiences gained from their parents and relatives. This underscores the importance of including the "significant others" in groups to target for behaviour change communication or other interventions aimed at improving cord care practices. Mothers that mentioned methylated spirit stated that they learnt it from the hospital while the knowledge of unhygienic substances

used was obtained from non-health professionals. It is noteworthy that none of the participants mentioned chlorhexidine as a recommended antiseptic for cord care. The poor knowledge about the use of chlorhexidine gel for cord care noted in this study is similar to reports from other Nigerian authors where its use by mothers was reported to be low [35–38]. The reasons proffered for the low use rate of chlorhexidine gel included the delay in cord separation time which discouraged its acceptability by some mothers [6, 38]. However, the World Health Organisation recommends daily chlorhexidine gel application to the umbilical cord stump in the early neonatal period for those born at home in settings with a high neonatal mortality (30 or more neonatal deaths per 1000 live births) [39]. Daily applications of 4% chlorhexidine gel for cord care during the first week of life has been shown to prevent several cases of newborn infections in low-income countries [20–23]. The non-awareness by the mothers in this study suggests that there is low knowledge of its usefulness generally by women in Bayelsa State. This calls for health education and engagement of relevant stakeholders to encourage its use as was done in Sokoto State where Orobatan *et al* [40] boosted the use of chlorhexidine gel in rural communities through active community involvement.

Even though more than half of the discussants at the FGDs attained a secondary level of education, their expected "good" knowledge does not seem to translate into good practices. This suggests that the level of literacy of the care providers is more important than that of the clients/patients in our study area. The three TBAs interviewed had no formal education and since the majority of the women in the community patronized them, little wonder then that the knowledge and practices of the TBAs held sway. The influence of TBAs on the knowledge and practices of cord care by mothers has been documented widely [7, 14, 16]. Lamawal *et al* [16] who interviewed thirty-one TBAs in Yenagoa LGA, Nigeria reported that only a few trained TBAs practiced good hand hygiene during cord care, most used black threads, plastic pegs and ropes to tie the umbilical cord and some used local herbs for cord care. As the major custodians of cultural practices in the community, the knowledge of cord care practices by TBAs could significantly influence the cord care practices of mothers in the community. Increasing the attendance of antenatal clinic by pregnant women and encouraging delivery by skilled birth attendants preferably in health facilities will improve women's knowledge and enhance appropriate cord care which will be initiated at the facility. Our study revealed the continuing significance of the TBA's and other places of maternal health care outside health facilities and these outlets should be included in programmes designed to improve maternal health and early child care.

## Conclusion

This study shows that myths, beliefs, and sociocultural practices still play significant roles in pregnancy, delivery and early child care particularly in the newborn cord care in Bayelsa State. Many pregnant women rely on inaccurate sources of information about cord care, patronize traditional birth attendants for antenatal care and are delivered at home by unskilled attendants, a situation that exposes the newborn to unwholesome practices including unhygienic cord care that can lead to infections and ultimately death. Behaviour change communication and public health programmes should be directed at improving antenatal care attendance and delivery at health facilities by pregnant women in both rural and urban communities in Bayelsa State. These interventions should also target the significant others who are often the source of knowledge and wrong practices adopted by mothers in the care of their newborn. Legislation that promote the adoption of good cord care practices should be advocated for at all levels of government, enacted and enforced. We recommend improved coverage of Bayelsa Health Insurance Scheme (BHIS)—the state's social health insurance scheme to make

healthcare delivery more accessible, affordable, and acceptable for pregnant women and their newborns wherever they live.

## Supporting information

**S1 Text. FGDs from participants from Ogbia LGA.**
(DOCX)

**S2 Text. FGDs from participants from Sagbama LGA.**
(DOCX)

**S3 Text. FGDs from participants from Yenagoa LGA.**
(DOCX)

**S4 Text. IDIs from participant from Ogbia LGA.**
(DOCX)

**S5 Text. IDIs from participant from Sagbama LGA.**
(DOC)

**S6 Text. IDIs from participant from Yenagoa LGA.**
(DOCX)

## Acknowledgments

We gratefully acknowledge our research assistants who assisted in the data collection and the study participants.

## Author Contributions

**Conceptualization:** Chika O. Duru.

**Formal analysis:** Adedotun D. Adesina.

**Funding acquisition:** Chika O. Duru.

**Project administration:** Chika O. Duru, Adedotun D. Adesina, Charles Tobin-West, Alice Nte.

**Supervision:** Abisoye S. Oyeyemi, Charles Tobin-West, Alice Nte.

**Visualization:** Adedotun D. Adesina.

**Writing – original draft:** Chika O. Duru.

**Writing – review & editing:** Ijeoma Nduka, Charles Tobin-West, Alice Nte.

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
