## [Decision Letter · Decision Letter 0]

25 Aug 2022

PGPH-D-22-01090

Socio-cultural practices, beliefs and myths surrounding newborn cord care in Bayelsa State, Nigeria: A qualitative study

Dear Dr. Duru,

Thank you for submitting your manuscript to PLOS Global Public Health. After careful consideration, we feel that it has merit but does not fully meet PLOS Global Public Health’s publication criteria as it currently stands. Therefore, we invite you to submit a revised version of the manuscript that addresses the points raised during the review process.

Thanks for submitting your manuscript for consideration. A minor revision is required prior to a final decision.

We look forward to receiving your revised manuscript.

Kind regards,

Charles Anawo Ameh, PhD

Academic Editor

Journal Requirements:

1. We ask that a manuscript source file is provided at Revision. Please upload your manuscript file as a .doc, .docx, .rtf or .tex.

2. In the online submission form, you indicated that "The data set for this study is available to be shared on request." All PLOS journals now require all data underlying the findings described in their manuscript to be freely available to other researchers, either 1. In a public repository, 2. Within the manuscript itself, or 3. Uploaded as supplementary information.

Additional Editor Comments (if provided):

Thanks for submitting your manuscript for consideration. A minor revision is required prior to a final decision.

Reviewers' comments:

Reviewer's Responses to Questions

**Comments to the Author**

1. Does this manuscript meet PLOS Global Public Health’s publication criteria? Is the manuscript technically sound, and do the data support the conclusions? The manuscript must describe methodologically and ethically rigorous research with conclusions that are appropriately drawn based on the data presented.

Reviewer #1: Yes

Reviewer #2: Yes

2. Has the statistical analysis been performed appropriately and rigorously?

Reviewer #1: N/A

Reviewer #2: N/A

3. Have the authors made all data underlying the findings in their manuscript fully available (please refer to the Data Availability Statement at the start of the manuscript PDF file)?

Reviewer #1: Yes

Reviewer #2: Yes

4. Is the manuscript presented in an intelligible fashion and written in standard English?

Reviewer #1: Yes

Reviewer #2: Yes

5. Review Comments to the Author

Reviewer #1: Congratulations on your paper! You have done a great work!

Understanding cultural beliefs about cord clamping and care requires background and social knowledge. Choosing to conduct a qualitative study was a good way to present these groups practices and how they are influenced by elder people. The use of threads, leafs, balms, toothpaste and ashes is commom in other poor countries, what explains the high rate of avoidable neonatal deaths. To interview and audiotape the sessions was clever. Minimizes comprehension issues the participants could have.

There is one major issue on supporting information session: the interviees said their names at the beginning of the meetings. You should supress this information, by using the designated identification written in the paper.

Minimal issues: some typos on discussion section.

Therefore, your paper is well written, refers to a commom problem in deprived places and employs a smart approach to access interviees opinions.

Reviewer #2: The paper is interesting. It highlights a topic that is very important.

Recommendations

Introduction

Add statistics concerning the complications related to the use of harmful practices. This is important to show evidence about the magnitude of the problem.

Situate a bit more the study in its context. Reflect on the healthcare system; maternal and child care; percentage of home deliveries and why; etc.

Mention the gap; what is missing.

Methods

Identify the criteria considered for the purposeful sample.

How many women in each FG?

Explain more the plan of analysis.

How did you ensure rigour?

Did you have to translate the language? If yes, how did you ensure accuracy of the meaning?

In the conclusion, add recommendations that make the healthcare system more accessible, affordable, and acceptable for pregnant families.

Edit English.

6. PLOS authors have the option to publish the peer review history of their article (what does this mean?). If published, this will include your full peer review and any attached files.

**Do you want your identity to be public for this peer review?** For information about this choice, including consent withdrawal, please see our Privacy Policy.

Reviewer #1: **Yes: **Roberta Maria P Azevedo

Reviewer #2: **Yes: **Mathilde Azar, PhD Health Sciences

---

## [Decision Letter · Decision Letter 1]

16 Feb 2023

Socio-cultural practices, beliefs and myths surrounding newborn cord care in Bayelsa State, Nigeria: A qualitative study

PGPH-D-22-01090R1

Dear Dr Duru,

We are pleased to inform you that your manuscript 'Socio-cultural practices, beliefs and myths surrounding newborn cord care in Bayelsa State, Nigeria: A qualitative study' has been provisionally accepted for publication in PLOS Global Public Health.

Best regards,

Melissa Morgan Medvedev, M.D., Ph.D.

Academic Editor

Reviewer Comments (if any, and for reference):

Reviewer's Responses to Questions

**Comments to the Author**

1. If the authors have adequately addressed your comments raised in a previous round of review and you feel that this manuscript is now acceptable for publication, you may indicate that here to bypass the “Comments to the Author” section, enter your conflict of interest statement in the “Confidential to Editor” section, and submit your "Accept" recommendation.

Reviewer #1: All comments have been addressed

Reviewer #2: All comments have been addressed

2. Does this manuscript meet PLOS Global Public Health’s publication criteria? Is the manuscript technically sound, and do the data support the conclusions? The manuscript must describe methodologically and ethically rigorous research with conclusions that are appropriately drawn based on the data presented.

Reviewer #1: Yes

Reviewer #2: Yes

3. Has the statistical analysis been performed appropriately and rigorously?

Reviewer #1: N/A

Reviewer #2: N/A

4. Have the authors made all data underlying the findings in their manuscript fully available (please refer to the Data Availability Statement at the start of the manuscript PDF file)?

Reviewer #1: Yes

Reviewer #2: Yes

5. Is the manuscript presented in an intelligible fashion and written in standard English?

Reviewer #1: Yes

Reviewer #2: Yes

6. Review Comments to the Author

Reviewer #1: Congratulations on your paper! It was a pleasure to re-review your study and notice how the suggestions added great value to it.

Still, I have some comments to share:

Introduction: I have added relevant information, what validates your study.

Methods: better structured and more detailed. It was easier to understand how the study was conducted.

Results: well written, clearer data and translated phrases. Also easier to understand.

Discussion: after the review it is easier to realize your train o f thought and how your results corroborate to literate.

Conclusion: you have added information about what could be done to improve the cord care, what enables people to understand why do we still have these practices nowadays. This may sound strange or unnecessary, but it is really relevant.

There are still some mispellings to correct, but most of the work is done.

Reviewer #2: No comments except that the rigour of the study is not explained.

7. PLOS authors have the option to publish the peer review history of their article (what does this mean?). If published, this will include your full peer review and any attached files.

**Do you want your identity to be public for this peer review?** For information about this choice, including consent withdrawal, please see our Privacy Policy.

Reviewer #1: **Yes: **Roberta Maria P Azevedo

Reviewer #2: **Yes: **Mathilde Azar
